# Relationship between Physicochemical, Techno-Functional and Health-Promoting Properties of Fiber-Rich Fruit and Vegetable By-Products and Their Enhancement by Emerging Technologies

**DOI:** 10.3390/foods12203720

**Published:** 2023-10-10

**Authors:** Alina Manthei, Gloria López-Gámez, Olga Martín-Belloso, Pedro Elez-Martínez, Robert Soliva-Fortuny

**Affiliations:** Department of Food Technology, Engineering and Science, University of Lleida/Agrotecnio-CeRCA Center, Av. Alcalde Rovira Roure, 191, 25198 Lleida, Spain; alina.manthei@udl.cat (A.M.);

**Keywords:** dietary fiber, non-thermal technologies, by-products, solubility, cholesterol, glucose reduction

## Abstract

The preparation and processing of fruits and vegetables produce high amounts of underutilized fractions, such as pomace and peel, which present a risk to the environment but constitute a valuable source of dietary fiber (DF) and bioactive compounds. The utilization of these fiber-rich products as functional food ingredients demands the application of treatments to improve their techno-functional properties, such as oil and water binding, and health-related properties, such as fermentability, adsorption, and retardation capacities of glucose, cholesterol, and bile acids. The enhancement of health-promoting properties is strongly connected with certain structural and techno-functional characteristics, such as the soluble DF content, presence of hydrophobic groups, and viscosity. Novel physical, environmentally friendly technologies, such as ultrasound (US), high-pressure processing (HPP), extrusion, and microwave, have been found to have higher potential than chemical and comminution techniques in causing desirable structural alterations of the DF network that lead to the improvement of techno-functionality and health promotion. The application of enzymes was related to higher soluble DF content, which might be associated with improved DF properties. Combined physical and enzymatic treatments can aid solubilization and modifications, but their benefit needs to be evaluated for each DF source and the desired outcome.

## 1. Introduction

The fruit and vegetable industry contributes significantly to global annual food waste on a weight basis (44%) [1,2]. Approximately 25% of the fruits and vegetables in the world are wasted post-harvest, mainly due to product grading to meet quality and acceptability standards, and during processing for juice and pulp extraction [2,3]. Juice production alone accounts for 5.5 million metric tonnes (MMT) of waste per year [3]. This waste contributes to climate change to a large extent when decomposing in landfills, emitting greenhouse gases and occupying critical resources, such as land, water, and energy [4]. Therefore, studies must be conducted to minimize post-harvest waste, as proposed in the Sustainable Development Goals of the United Nations, by developing sustainable technologies to enable the utilization of these by-products. The primary focus of interest lies in their application in food products as fiber enrichment due to their high content of dietary fiber (DF). Although the beneficial health effect of DF is widely known, the average intake of most European countries and others, including the USA, Australia, and New Zealand, does not meet the recommended intake of 25–35 g/d for adults [5]. Hence, the incorporation of fruit and vegetable by-products into food products would not only overcome potential environmental problems but also help to close the gap between actual and recommended DF intake in the population and improve human health.

As an introductory part, this work presents the properties of DF with a focus on its health benefits (Section 1.1); fruit and vegetable by-products that exhibit potential as innovative DF sources (Section 1.2); and physical and enzymatic treatments, as well as their combination, which were utilized to improve the characteristics of different fiber-rich sources (Section 1.3). Subsequently, the complex interaction of the physicochemical, techno-functional, and health-promoting properties of DF, particularly the antidiabetic potential, hypocholesterolemic effect, and fermentability, are described (Section 2). This knowledge assists comprehension of the third section of this work. Therein, several studies were selected that applied these treatments on novel DF substrates, explored the effect on several properties of DF, and thereby established connections between changes in structural, techno-functional, and health-promoting aspects (Section 3). Based on these findings, conclusions were drawn about the complex relationship between these DF characteristics and about the importance of certain structural characteristics for techno-functionality and health promotion. Simultaneously, a comparative analysis of studies that employed the same modification technology was conducted to evaluate the potential of each treatment for specific modulations. These comparisons could aid in identifying the ideal combination of DF source and technology for future applications to achieve the desired outcome. Lastly, an investigation is presented that explored glucose and cholesterol reduction of DF through in vitro experiments and validated the outcomes in vivo, with the objective of determining the feasibility of translating the health-promoting effect observed in vitro to an in vivo system (Section 4).

### 1.1. Composition and Health Benefit of DF

Fruit and vegetable by-products are rich in DF, which consists of plant carbohydrate polymers. These polymers encompass oligo- and polysaccharides (e.g., cellulose, pectin, hemicellulose, resistant starch, lignin), often linked with non-carbohydrate compounds, and are not digestible or absorbable in the small intestine [6]. DF is composed of a major insoluble part (IDF), primarily cellulose but also lignin, and a water-soluble fraction (SDF), comprised of some hemicellulosic but mainly pectic substances. The classification into these fractions depends mainly on the degree of DF polymerization and regularity of the backbone and side chains [7]. Besides DF, fruit and vegetable wastes are a source of phytochemicals, minerals, pigments, and organic acids [3]. Fruit, in particular, exhibits a high content of antioxidant compounds, mainly polyphenols and carotenoids, which are bound to DF and contribute highly to its health-promoting effect, such as the reduction of the prevalence of noncommunicable diseases (i.e., cancer, cardio-vascular and neurodegenerative diseases) [8,9,10]. However, solubility limits the capacity of DF to be integrated into food formulations. At least 30% of the total DF should be soluble when adding as a functional ingredient regarding nutritional, technological, and sensorial aspects [11]. DF high in soluble polysaccharides and polyphenols exhibits enhanced fermentability and ability to reduce blood glucose and cholesterol [7]. Furthermore, high solubility of a fiber-rich substrate is associated with favorable technological functionality, such as the capacity of forming gels, viscosity increase, emulsion stability, and water/oil binding capacity [8], as well as improved sensorial properties, such as reduced graininess and chewiness [12].

The non-digestibility of DF leads to its fermentation by intestinal microbiota inside the colon. During the metabolism of DF by gut bacteria, short-chain fatty acids (SCFA) (e.g., butyrate, propionate, acetate) are produced, leading to colonic acidification, which inhibits the growth of pathogenic organisms and carcinogens and thereby reduces the risk of colon carcinogenesis [13]. Additionally, DF as a prebiotic favors the growth of beneficial gut bacteria, which increase microbial community diversity and immune function of the host [14]. Besides fermentability, the other beneficial effect of regular consumption of DF on health is strongly based on its ability to bind different chemical compounds, enhance their excretion, and hinder or slow down their absorption in the gastrointestinal tract, particularly cholesterol, bile acids, and glucose. A high cholesterol blood level is associated with a high risk for hypertension, coronary heart diseases (CHD), stroke, and type 2 diabetes [15,16]; thus, the reduction of cholesterol by the intake of DF is of high interest. However, cholesterol is an important structural component of cell membranes, neurons’ myelin sheaths, initial substance of bile acids, vitamin D, and steroid hormones [16]. Bile acids are an essential part of the fatty acid digestion process, forming micelles, entrapping fat-soluble substances, and enabling their solubilization, transport, and uptake by the enterocytes in the duodenum [15,17]. Primary bile acids are synthesized by cholesterol from the liver. The terminal side chain can be conjugated with the amino acids taurine or glycine. When released into the small intestine, structural modifications by gut microbiota (deconjugation and 7α-dehydroxylation) of the bile acids, which escaped intestinal absorption, can occur, leading to the formation of secondary bile acids, such as deoxycholic and lithocholic acid [18]. Enterohepatic circulation leads to the reabsorption of more than 90% of the bile acids along the intestine to the liver. Only small amounts are excreted and are compensated by hepatic synthesis from cholesterol to maintain a constant level of primary and secondary bile acids [19]. Thus, not only the impeded adsorption and enhanced excretion of low-density lipoprotein (LDL)-cholesterol but also of bile acids account for the hypocholesterolemic and hypolipidemic effect.

Of equal importance is the binding of glucose, resulting in a decreased rate of intestinal absorbed glucose, which lowers the postprandial glycemic response and the subsequent production of insulin by the pancreas [15]. A reduced blood glucose level contributes to a prolonged time of feeling sated, improves insulin sensitivity, and reduces the risk of type 2 diabetes and increased body weight. Studies have indicated that hyperglycemia, resulting from insufficient or inefficient insulin secretion, not only interferes with the metabolism of macronutrients but can also induce non-enzymatic glycosylation of various proteins and thereby the development of chronic vascular complications in diabetes [20]. Hence, factors to optimize the hypoglycemic effect of DF need to be investigated and are important for the development of food products for diabetic patients.

### 1.2. Fruit and Vegetable By-Products with High Potential to Be Used as Novel DF Sources

When selecting fruit and vegetable by-products, primarily obtained after juice preparation and processing, as novel sources of DF, several factors should be considered. These include the quantity of generated by-products and thus their subsequent availability for utilization, which was assessed by the global production of the specific fruit or vegetable per year and the proportion of fruit that remains as by-product after processing. In addition, the selection takes into account the content of SDF and polyphenols (TPC) in the by-products since these structural elements exert a high impact on techno-functionality and the potential health-promoting effect. Based on these criteria, Table 1 summarizes fruits and vegetables and their by-products that might exhibit high potential to be used as innovative ingredients. By-products are mainly derived from peel or pomace/bagasse, which includes the remaining pulp, peel/skin, and seeds. Their content of SDF and bioactive compounds depend on different factors: fruit and vegetable varieties, harvest time, cultivation conditions, extraction method, and the preparation of the fiber source material. Generally, fruit contains a higher amount of pectin and bioactive compounds than vegetables, predominantly found in their peels [8,21]. Since some pectins change from insoluble to water-soluble during the ripening process, ripened fruits provide a higher amount of soluble polysaccharides and should be used preferentially [21].

The consumption of orange and apple juice is favored worldwide, resulting in the highest production volume but also the highest generation of residues (Table 1) [40]. Depending on the technology used for the juice extraction process, 20–40% of apple pomace is obtained as by-product [9], and about 50% of an orange fruit can be used as a potential DF source [41]. Considering also the high SDF content, including the high availability of pectic polysaccharides (17.31% in apple and 12.38% in orange pomace [9]) and their high phenolic content, both can be selected as sources with high potential. Additionally, citrus lemon and lime pomace constitute good DF sources, specifically regarding their high proportions of soluble polysaccharides (27.91% in lime, 28.10% in lemon) and high amounts of free and bound phenolic acids and flavonoids in their peels [42]. Lime peel and apple pomace are mainly used for the production of industrial pectin, mainly due to the abundant availability of the raw material and enhanced accessibility of pectin with high molecular weight, high degree of esterification, and good gelling properties [43]. However, mango peel and peach pomace show high potential to be used as additional sources. The main polysaccharides in the cell wall of peach fruits have been identified as pectin-like polymers [44], and a screening of mango peels demonstrated their high pectin content (up to 21.2%) and degree of esterification (ranging from 56 to 66%) [45]. Mango and peach deserve further attention also due to their considerable amounts of bioactive compounds, such as phenolics, carotenoids, and vitamin C [32,35].

Regarding the generation of vegetable residues, the high production of onions leads to elevated quantities of skins, roots, and discarded bulbs that did not meet the quality criteria [10]. DF composition differs depending on the layer of onions. Total dietary fiber (TDF) content increases through the layers from the bottom to the skin, whereas SDF decreases [38]. The outer, papery layers contain small amounts of phenolic acids, whereas the inmost layer exhibits significant quantities of flavonoids [46]. Thus, the inner bottom tissue might constitute a more promising source regarding the expected health benefit. However, the waste amount and TDF content are lower than those obtained from other fruit and vegetable sources. Another vegetable by-product with favorable properties is carrot pomace. After the extraction of carrot juice, up to 50% of the carrot remains as unused by-product [47]. The remnants contain a higher proportion of soluble DF components and amounts of polyphenols (e.g., phenolic acids and flavonoids) and carotenoids (e.g., β-carotene) than other vegetables, such as tomato, cauliflower, and potato [36,37]. The composition of the bioactive compounds varies greatly between the different cultivars and colors, where carotenoids appear in higher concentrations in the orange varieties and phenolics in the purple-yellow and purple-orange carrots [37,48].

### 1.3. Technologies with High Potential to Be Applied on Novel DF Sources

To alter DF composition, chemical, thermal, enzymatic, physical, and thermal-mechanical (e.g., extrusion, controlled instantaneous decompression) methods can be applied [6,49]. Before extracting/treating, preparation of the primary material can include wet milling, a washing step to remove unwanted substances; dehydration, such as tunnel, convection, or freeze drying; and dry milling; storage temperature is kept at −20 °C to prevent samples from microbial spoilage [10]. Parameters of the preparation method, such as the drying temperature and the particle size obtained after milling, can already affect the characteristics of the DF product. Therefore, it is crucial to carefully select these parameters to ensure the desired qualities and microbial safety. When applying modification techniques, the objective is to obtain a DF ingredient with improved solubility, techno-functionality, and health impact, and to maximize the extraction yield but minimize the loss of bioactive heat-labile compounds. In particular, the proportion of SDF is of great importance not only for the health benefit but also for food processing and product development. Thus, it is essential to explore techniques specifically aimed at increasing the solubility of DF derived from fruit and vegetable by-products.

Conventional techniques include thermal (e.g., pressure cooking, boiling, and sterilization), chemical (e.g., addition of hydrochloric acid, sodium hydroxide, alkaline hydrogen peroxide, or carboxymethylation treatment), and well-established physical procedures (i.e., grinding, high-speed mixing, and steam explosion). Generally, these methods can alter the plant cell wall composition and enhance DF solubility by disrupting glycosidic bonds and reducing cohesion among cell wall polysaccharides through high temperature, chemical hydrolysis, and mechanical shear [6]. However, these traditional approaches have some disadvantages, including high energy and time consumption, potential damage of DF structure, degradation of bioactive compounds, and the critical environmental impact of chemical solvents. Therefore, the development of novel strategies to maximize the extraction yield while maintaining or improving the functionality of DF in an environmentally friendly way is essential for achieving an efficient exploitation of food by-products. Non-thermal technologies, such as ultrasound (US), pulsed electric fields (PEF), and high-pressure processing (HPP), and innovative heating technologies such as microwave and extrusion, have shown potential as alternatives to traditional methods for extracting DF. They are energy-efficient and environmentally friendly technologies since their application decreases processing times, use of toxic solvents, and achieved temperature, while preserving heat-sensitive nutrients. In principle, most of these treatments use high mechanical shear forces and/or high local pressure and temperature, which can modify and break inter- and intramolecular polysaccharide bonds and thereby reduce the size of suspended particles, modulate the DF surface structure to a more porous state, transform some IDF components to SDF, and open the DF structure, resulting in the enhanced availability of certain functional groups within the inner structure of DF [6]. Several studies have demonstrated that they are suitable for modifying the molecular structure of DF and releasing bioactive components, enhancing the antioxidant activity. A combinations of methods, such as enzymatic or microbial pre- or post-treatment, can be even more effective to change the molecular structure and enhance DF solubility [50].

US can be described as “high frequency vibration that generates fluid mixing and shear forces on a microscale” [51]. Ultrasound waves contain high- and low-pressure regions, resulting in fluid motion (acoustic streaming) and shear forces due to the cavitational collapse of bubbles. Collapse occurs when a bubble or a cloud of bubbles grow by rectified diffusion (entry of gas into the bubble) or coalescence (association and merge of bubbles) and reach a certain size [52]. This microscale implosion induces high shear forces, localized increase of temperature and pressure, and the generation of hydroxyl radicals, which can lead to the breakage of cell wall polysaccharides and their degradation to shorter DF fractions. Additionally, the process facilitates the release of intracellular molecules, such as bioactive compounds [51,53]. The use of high intensity (10–1000 W cm^−2^) and low frequency (16–100 kHz) is suitable for modifying the physicochemical properties of DF [54]. Nevertheless, optimization of the conditions to achieve maximal yield and desired health-promoting or technological properties needs to be conducted in terms of frequency, energy applied, propagation of cycle, treatment time, and achieved temperature [3].

The application of PEF is another rapidly growing research field in the food processing industry. This non-thermal method uses short duration pulses between two adversely charged electrodes, initiating a transmembrane potential. When a critical value is reached, repulsion between charged molecules causes the formation of pores and increases the cell permeability (electroporation) [55]. Pulses of moderate electric field strengths (0.5–10 kV cm^−1^) with relatively low energy input (0.1–2 kJ kg^−1^) can induce a stress response in plant tissues, resulting in increased permeabilization and structural modifications of macromolecules [56]. The main influencing parameters, which need to be finely tuned, are energy input, electric field strength, frequency, pulse shape and number, and achieved temperature [3]. The optimization of these parameters could enhance the extraction of functional ingredients and intracellular compounds.

Similarly, applying high pressure, such as high hydrostatic pressure (HHP) or high-pressure homogenization (HPH), can cause damage to cell membranes, increased cell permeability, and the formation of smaller particles [54]. The principle of HHP is based on the application of uniform and instantaneous ultra-high pressure (100–600 MPa) on a hermetically sealed food product within a thermally insulated airtight vessel, which primarily results in the breakdown of hydrogen bonds [57]. Pressure is transmitted by a liquid medium, normally water [58]. On the other hand, during HPH, a fluid is forced through the narrow gap of a homogenization valve by employing ultra-high pressure. Within the valve, cavitation, turbulence, and hydraulic shear between the fluid and valve seat act on the fluid and cause the break-up of particles and change of molecular structure [59]. In particular, HPH was reported to produce smaller particle sizes than conventional homogenization techniques due to the higher possible pressure range [60]. Besides the applied temperature and treatment time, the applied pressure delivered from the high-intensity pump has the highest impact on the outcome of high-pressure treatments.

Microwave is an innovative thermal technology that uses an electromagnetic field, containing two perpendicular fields (i.e., magnetic and electric field) from 300 MHz to 300 GHz. Microwave energy is converted into heat through dipole rotation and ionic conduction, which causes, supported by increased pressure, separation and dissolution of molecules from the matrix [3,61]. Hence, microwave treatments have the potential to solubilize fiber components. Since microwave energy is directly dissipated to the reaction components, high heating rates are caused, and the sample inner temperature can rise rapidly [62]; thus, experiments with molecular structures containing heat-labile compounds require the optimization of treatment time and energy to accurately control the temperature.

Extrusion is a thermal-mechanical process combining high temperature, pressure, and shear forces when pushing the material through a narrow opening (die). When the product leaves the extruder, the sudden decrease in pressure converts water to steam resulting in an expansion of the material and modification of the spatial inter- and intramolecular DF structure [57,63]. The outcome is highly dependent on the conditions applied. Severe conditions include high temperatures (>200 °C) and low moisture content (<15%), whereas mild conditions apply high moisture content (>15%), low residence time (0.5–1 min), and lower temperatures (<180 °C) [64]. Extrusion cooking with high temperature, pressure, screw speed, and moisture content showed effectiveness in changing the chemical nature of polysaccharides and increased the solubility of DF [63]. However, the content of bioactive compounds tends to decrease under these severe conditions [64]. Several other parameters impact the process, such as die shape, pressure, number of screws (single- or twin-screw), screw speed, feed moisture, and feed ratio, and need to be selected according to the material.

Enzymatic hydrolysis can be applied as a supporting treatment prior to a physical modification in order to loosen the fiber network structure or subsequently when the access for enzymes to certain linkages inside the fiber structure is facilitated [65,66]. Enzymes can effectively depolymerize and solubilize fibers due to their high specificity of hydrolyzing certain linkages of different fiber components. Cleavage of the linkages of the different fiber components requires the use of different kind of enzymes, primarily cellulases, hemicellulases/xylanases, and pectinases. Cellulases, mainly composed of β-glucanases and β-glucosidases, specifically hydrolyze the (1,4)β-D glucosidic bonds in cellulose. Due to the heterogeneous composition of hemicellulose, an effective hydrolysis of this DF component requires the use of an enzyme preparation containing different enzyme activities (e.g., β-xylosidase, endo- and exoxylanases) [57]. Most abundant among the pectinolytic enzymes are polygalacturonases, which catalyze the hydrolysis of (1,4)α-D galactosiduronic linkages in pectates, targeting the backbone of pectin. The combination of enzymes has the potential to increase the efficiency of the enzymatic treatment by synergistic effects among the enzymes. These synergistic effects can imply the reduction of steric hindrance resulting from the removal of interfering substituents by a debranching enzyme, such as pectinase, and thus providing access for depolymerizing enzymes (i.e., cellulase) [67]. By combining a physical modification technique with an enzymatic treatment, their effect on the modification of the DF microstructure and the release of shortened soluble fractions can be optimized and limitations of both technologies can be overcome.

## 2. Impact of Physicochemical Properties of DF on Its Techno-Functional and Health-Promoting Characteristics

Binding capacities and the cholesterol- and glucose-lowering effect of DF are the result of a complex interaction of structural and physicochemical properties (i.e., particle size, surface characteristics, and DF composition, including solubility and total phenolic content) and techno-functional properties (i.e., water- and oil-binding, viscosity, and cation exchange capacity) [7]. Hence, certain parameters, such as hydration properties and the presence of different functional groups, can serve as indicators to estimate a high or low effect, as shown in Figure 1.

### 2.1. Relationship between Physicochemical and Techno-Functional Properties of DF

Water retention and oil-holding capacities are important techno-functional characteristics for product development since they confer to DF the ability to stabilize emulsions, replacing fat, flour, or sugar, modifying the texture and sensorial properties of food products, and reducing syneresis (the separation of a liquid from a gel caused by contraction), which leads to improved moisture retention and storage stability [6,50]. Functionality is affected by certain structural properties, such as the soluble content, hydrophilicity, hydrophobicity, and cellulose crystallinity.

Water retention capacity (WRC) accounts for fiber-swelling, gel-forming, and thickening capacities and is based on two mechanisms: (1) Water can be bound to hydrophilic groups of DF, such as hydroxyl, carbonyl, and carboxyl groups, by polar interactions and hydrogen bonding. (2) Adjacent polymer strands of DF can combine in ordered assemblies (junction zones), leading to the formation of a three-dimensional network wherein large amounts of water can be entrapped [68]. SDF is composed of a large number of hydrophilic groups and has a high molecular weight and branched structure, which impart this fraction high hydration capacity and viscosity [6,7,69]. The presence of hydrophobic groups, and thus lipophilic sites, primarily accounts for the oil-holding capacity (OHC). Other structural characteristics that can contribute to an improved OHC are a high charge density, lignin, and protein content of the polymer [66,70,71]. At the same time, high WRC and OHC are linked with improved emulsifying properties of DF, mainly due to the stabilization of the emulsion gel matrix structure [70].

The major DF component is cellulose, consisting of 70% orderly crystalline regions and 30% amorphous regions [72]. Crystalline cellulose is formed by hydrogen bonding and van der Waals forces between adjacent molecules, resulting in a dense and packed structure. On the other hand, amorphous regions contain mainly non-crystalline cellulose, hemicellulose, and lignin [73]. A low crystallinity index, indicating a decreased proportion of dense, crystalline cellulose, correlates with higher porosity; stronger, entangled gel networks; and higher WRC. In contrast, increased cellulose crystallinity is related to enhanced thermal stability and OHC but lower hydration [74].

### 2.2. Impact of Physicochemical and Techno-Functional Properties of DF on Its Health-Promoting Effect

The health-promoting effect of DF is highly connected with its ability to reduce blood cholesterol and glucose. This reduction is based on: (1) the physicochemical entrapment of the compounds inside the DF network; (2) the adsorption of these substances by their direct association with functional groups of DF. Both mechanisms, entrapment and adsorption, are responsible for the retardation and thereby the delay of diffusion of the compounds from the lumen to gut epithelial cells [21]. The main factors impacting adsorption and retardation are the DF composition, structure of the component and the environment, including pH, ionic strength, temperature and duration of exposure [49].

As the preferential mechanism, the compounds are captured and entrapped by the viscous polymer network based on the gel-forming properties of DF [75]. The soluble fraction is associated with a higher ability to form gels and increase viscosity linked to its high molecular weight and entangled conformations [6]. Hence, higher solubility, indicated by smaller particle size and higher surface area, provides stronger, more viscous gels and ideal structural conditions to entrap the compounds and reduce their diffusion [76,77]. As mentioned above, when hydrophilic groups are exposed and/or crystallinity is decreased, WRC is enhanced, which facilitates the capturing effect inside the DF network. However, the IDF fraction constitutes a physical obstacle and should be considered as an influencing but secondary factor when evaluating the effect of DF on the delay of diffusion [69].

Direct interactions, primarily non-covalent chemical bonding, such as hydrogen bonds, hydrophobic interactions, van der Waals forces and electrostatic interactions, require the availability and exposure of side chains and functional groups [78]. The major components of DF, namely cellulose, hemicellulose and pectin, provide a high number of hydroxyl and carboxyl groups favouring hydrogen bonding. However, apolar surfaces can be generated depending on the monomer ring conformation, stereochemistry of the glycosidic linkages and the degree of hydration and amount of intra-molecular hydrogen bonding [7]. The presence of non-polar molecules, such as hydrophobic aromatic rings of phenolic compounds and carotenoids, also increase hydrophobicity and probability for interactions. Additionally, DF contains charged polysaccharides, such as carboxyl groups from pectin, which play an important role for mineral absorption [49]. By changing pH and ionic strength, more charges are produced, and components can be additionally retained by electrostatic interactions.

#### 2.2.1. Antidiabetic Potential

Determination of glucose adsorption and retardation capacities (GAC/GRC) serve as parameters to evaluate the in vitro antidiabetic potential of DF. Direct interactions between DF and glucose molecules can be assigned to polar and non-polar groups [79] whereas dipole-dipole interactions might be the primary non-covalent bonding type due to the high polarity of glucose. Several studies have reported positive correlations between porosity, surface area and the soluble content on GAC and GRC [65,80]. For instance, Huang et al. [80] compared the delay of diffusion and glucose adsorption of fiber-rich orange pomace, cellulose and psyllium containing a high soluble DF content. The highest GRC and GAC were found for psyllium, which was attributed to the higher number of soluble fibers. Therefore, the increase of the soluble content within a fiber system containing insoluble and soluble fraction might be favourable particularly to form a more viscous network structure and entrap glucose molecules.

The reduction of blood glucose level and the risk of suffering type 2 diabetes are also influenced by the adsorption and inhibition of α-amylase (AAIR) by DF components. DF impedes the accessibility to the enzyme substrate and, consequently, the inhibition of starch degradation to glucose [81]. The main mechanisms behind this inhibition are entrapping the enzyme and starch inside of the fiber matrix, and the adsorption of α-amylase and starch by DF components which leads to a reduced contact rate and hydrolysis. An enhanced SDF content and viscosity facilitate embedding the compounds and reducing their accessibility [70,72]. In contrast, insoluble substances, mainly cellulose, are involved in adsorption [36,82], which is affected by its crystallinity index.

#### 2.2.2. Hypocholesterolemic Effect

Cholesterol reduction is based on two mechanisms: the adsorption and increment in the excretion of cholesterol and, secondly, the adsorption and enhanced excretion of bile acids, preventing their reabsorption by the liver and promoting the synthesis of new bile salts from cholesterol [83]. Suitable parameters to predict the hypocholesterolemic effect in vitro are adsorption and retardation capacities of bile acids (BAC, BRC), the adsorption capacity of cholesterol (CAC) and the cation exchange capacity (CEC). Bile acids have a steroid nucleus and an aliphatic side chain, thus contain a hydrophobic and hydrophilic surface enabling them to form micelles above a certain concentration (CMC) and interact with an oil-water interface. Similarly, cholesterol shows amphiphilic nature since it is comprised of an apolar ring system, which is associated with a hydrocarbon chain and a hydroxyl group [16].

The main mechanism causing the enhanced excretion of cholesterol is due to the physical entrapment of bile acids and cholesterol and their micelles, promoted by high SDF content and increased viscosity [7,83,84]. Additionally, SDF is discussed to support the physical barrier properties of the unstirred water layer, which covers the luminal side of the enterocytes, and therefore impair the uptake of bile salt-cholesterol micelles [17]. Soluble carboxymethyl cellulose (CMC) is discussed as the main soluble component of DF which can entrap cholesterol crystals by forming cholesterol-CMC-composites [85]. However, a study with heat damaged oat fiber with decreased viscosity did not show the expected decrease of BAC and indicated that not solely viscosity and solubility impact cholesterol lowering but direct binding of the components [84].

The mechanism of the adsorption of bile salts and cholesterol by DF is still unclear [83,84]. Research is mainly focused on studying the interaction between bile acids and SDF since it is expected to be the main process of cholesterol reduction besides entrapment. Bile acids were better absorbed when DF had a hydrophobic surface suggesting that hydrophobic interaction is the predominant non-covalent binding type [7]. This might explain the correlation between the ability to bind bile acids and to retain fat linked by the hydrophobic nature of DF [71]. However, the relationship between OHC and BAC is also determined by SDF since certain soluble components, including pectin, arabinoxylan and arabinogalactan, have high affinity for lipid materials [86]. Hydrophobic surface of DF or rather potential binding sites can be enlarged by the presence of free hydrophobic groups from polyphenols [7]. Different studies confirmed that the presence of polyphenolic compounds, such as phenolic acids and flavonoids, and lignin, contribute to BAC through hydrophobic interactions [87,88]. Other factors, such as ionic strength, might as well have an impact since it promotes electrostatic interactions, but studies are lacking [21,49]. Contrasting, cholesterol was suggested to be preferably bound to insoluble hemicellulosic and cellulosic components. For instance, coffee parchment samples high in IDF, composed by hemicellulosic and cellulosic profile, exhibited high CAC [89]. However, direct interactions of IDF with cholesterol have a lower contribution to the hypocholesterolemic effect than bile acid binding by hydrophobic attractions [84].

Cation exchange capacity (CEC) is considered as a techno-functional property since it measures the ability of DF to retain cations on the surface or rather the amount of cation that can be exchanged by another. DF with high CEC can contribute to the destabilization and disintegration of micelles when forming fiber-micelle complexes which act as barriers and reduce the diffusion and adsorption of lipids and cholesterol [90]. Hydroxyl and carboxyl groups of polyphenols, lignin and from uronic acids of the pectin and hemicellulose fraction (i.e., glucuronoxylan) have exchange ability [49,91]. Hence, the contents of polyphenols, lignin and uronic acid of DF should be taken into account as contributing factors to impact CEC and the reduction of cholesterol.

#### 2.2.3. Fermentability

Fermentability of DF is mainly measured by the in vitro production of gas and SCFA as main metabolites, measurement of pH and bacterial composition during batch or continuous fermentation. The uptake and metabolization of DF by gut microbiota species are strongly connected with DF structure whereas soluble DF is more easily fermentable than insoluble DF [6]. This is mainly due to the lower degradability of insoluble DF for bacterial polysaccharide hydrolases based on their lower accessible, more dense and cross-linked cell wall structure [92] and the enhanced metabolization of shortened DF fractions of low DP, such as cello- xylo and pectin oligosaccharides [93]. Additionally, the utilization of carbohydrates by microbiota is strain-dependent since the bacterial genome encodes enzymes that hydrolyze carbohydrate linkages, such as glycosyl hydrolases (GH), and other protein degrading enzymes responsible for carbohydrate-binding and transport [94]. Studies are still scarce investigating the relationship between different DF structures and fermentation profiles. Some studies suggest that not only the DF composition, such as monosaccharides, but also the chain structure, such as linkages and oligosaccharide units, play a key role in the utilization of DF and production of metabolites [94]. In addition, the presence of polyphenolic components, which are highly present in fruit by-products, has been linked to the inhibited growth of some pathogenic species, such as *H. pylori*, but also the growth of several beneficial bacteria strains. As a result, DF sources containing a high TPC promote a positive balance in the gut microbiota composition and thereby show enhanced fermentability [95].

## 3. Enhancing the Health-Promoting Effect of DF by Applying Novel Technologies

As discussed in Section 2.2.1 and Section 2.2.2, DF with a high soluble content and hydrophobicity, promoted by the presence of certain groups, such as polyphenols and lignin, and hydrophobic surface configuration, are likely to show a high hypocholesterolemic effect and antidiabetic potential. The application of processing technologies can lead to this desired structural configuration and promote these health-promoting properties (Table 2). Generally, such technologies reduce the particle size and induce structural alterations, such as a more porous surface structure and the transformation from IDF to SDF. Furthermore, they can cause the opening of the entangled DF structure and the exposure of certain functional groups that improve the adsorption and retardation capacities of glucose, bile acids, and cholesterol (Figure 1) [6]. The selection of the processing technology and treatment conditions need to be adapted to the specific DF material to avoid detrimental structural changes. For instance, processing by ultrafine grinding/comminution of different pomaces, such as citrus IDF [96], grape [97], pear [98], and carrot IDF [79], could induce the solubilization of certain components, detected by a high increase of the SDF proportion. However, techno-functionality (i.e., WRC and OHC) decreased. Hence, mechanical comminution to a very low particle size (D_50_: 9.1–38.4 µm; D_90_: 31.6–115.2 µm) is advantageous to obtain a high SDF proportion but likely to cause structural collapse and the decrease of water- and oil-binding properties (Figure 2). Therefore, in order to avoid reductions in WRC and OHC, it is important to optimize the particle size of the pomace powder and the ratio between IDF and SDF. Results of the effect of ultrafine grinding techniques on the health-related properties are limited, but the available data suggest that the intense destruction of the polysaccharide chains does not have an adverse effect on GAC and CAC but might promote binding of glucose and cholesterol. The study of Peerajit et al. [76] showed that the particle size reduction of DF powder by grinding from lime residues did not influence GAC, regardless of the glucose concentration, but significantly improved GRC and BRC. In conclusion, a very low particle size together with higher porosity and solubility can likely deteriorate techno-functional properties but might enhance the capturing effect of DF and facilitate the embedding of certain molecules inside of the loosened network.

Generally, using chemicals as a traditional technology to improve DF properties is associated with a risk of causing structural damage and impairing the functionality and health-promoting properties of DF. For instance, acidic treatment of coconut cake DF (soaking in 1 M NaOH) led to reduced total and soluble DF percentages and WRC, which was associated with decreased BAC and AAIR. The removal of amorphous components, such as hemicellulose and starch molecules, increased the crystallinity index and reduced hydration capacity [70,75]. Chemical treatments are less selective than enzymatic hydrolysis, which make them more likely to cause damage of the structural framework. The application of alkaline treatment (soaking in 0.5 M NaOH) to extract DF from deoiled cumin seeds was less effective than enzymatic treatment, and shear emulsifying assisted enzymatic hydrolysis. Chemical-treated DF exhibited lower functionality (WRC and OHC) and physiological values, namely BRC, GAC, and AAIR [72]. However, alkaline hydrogen peroxide treatment of citrus peel (mixing in 1% hydrogen peroxide solution at pH 11.5) caused better hydration properties than homogenization [100]. In summary, chemical treatments can be applied to soften the structure and potentially achieve better techno-functional properties but, regarding the environmental difficulties and consumer concerns related with the use of acidic chemicals during production, they should not be the preferential method.

Enzymatic, US, extrusion, and high-pressure treatments have shown promising results for enhancing the health-promoting properties of DF. Several studies on different DF sources confirm the high efficiency of enzymes to increase the SDF content, but the results regarding the impact on techno-functional properties are not consistent (Table 2). However, in most of the studies, higher SDF content induced by enzymatic hydrolysis led to the promotion of health-related adsorption and retardation capacities [65,75,79,101]. When different treatments were applied, physical treatments (such as US on tea seed IDF [65], HPH on DF from bamboo shoot shell [101] or on pear and IDF carrot pomace [79,98]) have shown more effectiveness than enzymes in causing structural modifications that are linked with high techno-functionality and glucose and cholesterol reduction (Figure 2). For instance, extrusion changed the molecular structure of orange pomace DF and the content of SDF, which enhanced WHC and its health-promoting potential, particularly BAC and AAIR, significantly [80,102]. Naumann et al. [103] found a linear correlation of the decrease of the permeability rate and diffusion of bile acids when the soluble proportion of lupin kernel fiber was increased by extrusion. Results achieved with HPH agree on its effectiveness to particularly improve WHC, which was associated with higher GAC and CAC (Table 2). Additionally, high-pressure application caused enhanced solubilization and extraction efficiency of polyphenols and carotenoids from carrot IDF [79] and tomato pomace [104], indicated by increased contents and antioxidant activity. TPC after US treatments of cashew apple bagasse [105], pomegranate peel [106], and black chokeberry waste [107] were also improved, and extraction times were reduced.

**Table 2 foods-12-03720-t002:** Treatments conducted on high-DF by-products and their impact on physicochemical, techno-functional, and physiological properties associated with hypocholesterolemic effect and antidiabetic potential; comparison of untreated (NT, first row) and treated product (second row).

DF Source	Treatment Conditions (Optimum)	Physico-Chemical	Techno-Functional		Hypocholesterolemic Effect		Antidiabetic Potential			Comments
		SDF [%]	WHC [g/g]	OHC [g/g]	BAC [%]	CAC [mg/g] pH2/pH7	GAC [mM/g]	GRC [%]	AAIR [%]	
**Ultrafine grinding**										
Carrot pomace IDF [79]	Ultrafine comminution to 40.05 µm	2.07 6.04	4.50 * 4.00 *	1.85 * 1.60 *		29/30 35/36	**100 mM** 1 2.3			Reduced WRC, OHC, TPC, and total antioxidant activity
Pear pomace [98]	Superfine grinding: micronizer, 25 min	10.0 13.7	3.44 2.73	1.82 1.09		1.27/3.88 3.40/5.64 g/mL				Destruction of polysaccharide chains with adverse effect on WRC and OHC
Citrus pomace IDF [96]	1. Micronizer, 8 min 2. Jet mill, 210 kW, 2100 r/min	8.10 14.40 14.90	7.33 6.44 5.74	1.34 1.13 1.28						Grinded DF with smoother surface but lower techno-functionality
**High-pressure processing (HPP)**										
Bamboo shoot shell IDF [101]	100 MPa, 5 circles		2.99 8.27	5.79 8.16		**pH 7** 8 * 12 *	**20 mM** 0.0038 0.0045			Honeycomb microstructure after HPH with enhanced properties
Peach pomace IDF [77]	120 MPa, 1 circle		5.02 7.65	2.44 7.58			**100 mM** 0.72 1.9	27.81 42.62	28.69 47.07	Enlarged surface area with higher glucose-lowering properties
Pear pomace [98]	300 MPa, 15 min	10.0 16.0	3.44 5.77	1.82 2.77		1.27/3.88 4.92/8.37 g/mL				Enhanced SDF content, leading to improved CAC; higher effect than grinding
**Ultrasonication (US)**										
Tea seed IDF [65]	640 W, 20 min, 40 °C		44.29 60.15	18.37 30.42		8.76/6.18 12.71/8.67	**0.5 mM** 0.059 0.073			Highest improvement of properties after US (better than EH)
Garlic straw IDF [108]	535 W, 41 min, 45 °C		7.38 9.72	8.96 10.56		10.59 16.75	**100 mM** 4.27 4.65		28.68 33.91	Honey-comb structure after US with higher porosity
**Microwave (MW)**										
Okara [109]	600 W, 2.5 min	2.24 7.69				8.80/18.46 9.18/18.09		6 * 20 *	9.55 20.92	Damage of crystalline structure, improved thermal stability
Apple pomace SDF [110]	Extraction of SDF by 1. Acid 2. Microwave: 80 °C, 2 min, 550 W		3.0 6.9 *	1.3 2.0 *						Enhanced SDF yield after microwave extraction
**Extrusion**										
Orange pomace [102]	129 °C (barrel T), 15% (feed moisture), 299 rpm (screw speed)	17.31 30.29	5.80 6.73	1.23 0.84	**Bile acid mixture** **pH 7** 38.5 61.1	**Buffered micellar solution** **pH 6.3** 6.89 11.92	**50 mM** 0.42 0.46 **100 mM** 0.720.75	19.25 21.12	22 * 38 *	All properties improved except from OHC, treated pomace with higher health-related properties than cellulose but lower than psyllium
Lupin kernel [103]	150 °C, 20% moisture, 400 rpm	1.9 29.3	7.71 15.04	2.28 1.25 *						No effect on molecular interaction (BAC) with chenodesoxcholic acid but improved BRC related to increased viscosity
**Enzymatic hydrolysis (EH)**										
Defatted coconut cake (unscreened samples) [70]	Cellulase, 1 h	19.33 30.00	13.08 8.39	9.91 4.42	**1 mg/mL SC** 70/19 75/20	79/45 82/46 %			10.52 28.82	Decrease of techno-functional properties of unscreened samples (particle size > 250 µm), enhanced SDF with enlarged surface area
Carrot pomace IDF [79]	Cellulase and xylanase, 2.5 h	2.07 15.07	4.50 * 5.75 *	1.85 * 2.00 *		29/30 37/38	**100 mM** 1 2.4			High increase of SDF but improvement of properties lower than with HPH
DF of deoiled cumin seeds (unsieved samples) [72]	1. Alkalic extraction: soaking in 0.5 M NaOH, 2 h 2. Enzymatic: 4.5% alcalase, 2.4 L, 155 min	0.42 7.85	3.3 5.48				**50 mM** 3.48 3.89		8.75 11.08	Improved BAC, BRC, and AAIR after EH assisted extraction
**Combination of treatments**										
Peach pomace SDF [111]	1. NT SDF: Water extraction 2. EH: 2% cellulase, 10 h 3. HPH + EH: 140 MPa, 4 cycles + 2% cellulase, 6 h	7.3 32.6 36.3			376.04 530.87 465.95 **mg/g**	7.57/14.75 19.21/30.05 25.31/26.94				HPH improved the recovery yield of SDF, and time of 10 h EH could be reduced to 6 h; enhancement of CAC with combined treatment
Tea seed IDF [65]	1. NT IDF 2. EH: cellulase and xylanase, 36 h 3. US + EH: 640 W, 20 min, 40 °C + cellulase and xylanase, 36 h		44.29 59.34 51.89	18.37 21.67 19.58		8.76/6.18 6.56/5.94 7.6/7.03	**0.5 mM** 0.057 0.063 0.061			No further improvement of properties with combination; highest improvement with sole US
Rose pomace IDF [73]	1. NT 2. EH: 230 U/g DF cellulase, 900 U/g DF xylanase, 2 h 3. US + EH: 150 W, 30 min + cellulase, xylanase, 2 h	4.45 15.41 13.59	8.32 10.45 11.09	3.13 4.47 4.87		12.01/13.24 12.79/13.42 13.98/14.50	**200 mM** 46.5 * 49.0 * 45.0 *			No further promotion of IDF to SDF conversion by coupled treatment but further enhancement of techno-functional properties
Seeds of Akebia trifoliata (Thunb.) Koidz. Fruits [112]	1. Alkalic extraction 2. EH: cellulase, 1 h 3. US + EH: 500 W, 40 °C + cellulase, 1 h 4. MW + EH: 500 W, 40 °C, 1 h + cellulase, 1 h	5.33 5.85 5.51 6.04	7.46 7.99 7.82 5.64	3.55 3.57 3.95 4.18	26.00 * 29.21 31.23 37.00	28/41 * 38/47 * 40/60 * 45/62 *	**0.1 M** 0.50 * 1.35 0.87 0.38 *		6.62 5.00 * 6.81 7.45	Further improvement of OHC, BAC, CAC, and AAIR after enzymatic extraction combined with physical pre-treatments

*** Approximate values from the graphs of the articles; SDF**—soluble DF content, **WHC/OHC**—water and oil holding capacity, **BAC/CAC**—bile acid and cholesterol adsorption capacity, **GAC/GRC**—glucose adsorption and retardation capacity, **AAIR**—α-amylase inhibition rate. Retardation capacities were evaluated after 1 h of dialysis. The solutions, which were used for the BAC, CAC, and GAC incubations, are stated since they differentiate between the studies; if no incubation solution for CAC is specified, a diluted egg yolk solution was used.

Some authors have studied the application of physical treatments together with enzymatic hydrolysis for enhancing DF functionality. Interestingly, the application of a combined treatment of US and subsequent enzymatic hydrolysis did not show further improvement of DF properties. Although the ultrasonic pre-treatment is expected to loosen and expand the DF structure by increasing the interspaces and thus facilitating access to the enzymatic points of attack inside of the DF matrix [113], the sole ultrasonic treatment of tea seed IDF exhibited the best functional and physiological properties [65]. Similarly, ultrasonic pre-treatment did not benefit the enzymatic-assisted extraction of DF from rose pomace and did not further increase SDF proportion or GAC [73]. In contrast, enzymatic hydrolysis and alkaline extraction were studied in combination with different physical pre-treatments to assist the DF extraction from Akebia trifoliata (Thunb.) Koidz. seeds. The pre-treatment with US improved AAIR, while microwave had a promoting effect on BAC [112]. Hence, mechanical-enzymatic procedures can enhance the extraction efficiency and nutritional effect of DF, but the benefit depends highly on the initial accessibility of DF components for the enzymes. The crystallinity index might serve as an indicator for the accessibility of cellulose since it reflects the proportion of amorphous regions, which are the enzymatic attacking points [114]. Hence, DF with high crystallinity index is likely to benefit from a physical pre-treatment, whereas DF with low crystallinity might not require the mechanical opening of the fiber matrix, and further solubilization by a subsequent enzymatic hydrolysis can lead to partial structural collapse of the insoluble matrix and thus impairment of techno-functionality.

Microwave and PEF applied on DF sources have been demonstrated to exert beneficial effects to preserve and optimize the content of bioactive compounds and antioxidant activity. For instance, applying microwave on mango peel increased tannin and proanthocyanidin extractability and achieved a six times higher antioxidant activity compared to conventional solvent extraction in a water bath [115]. Corrales et al. [116] tested PEF, US, and HHP for extracting bioactive compounds from grape by-products (skins, stems, and seeds), and the highest increase among the different technologies of polyphenols were found with the PEF treatments. However, studies of these technologies on altering DF properties are scarce. The application of microwave on okara DF demonstrated the promotion of its solubility and glucose-reducing properties [109]. Promising results were reported in PEF-treated pectin from sugar beet pulp. Treatments modified its physicochemical properties (i.e., decrease of the molecular weight, esterification, and increase of available carboxyl groups) [117]. Considering the efficacy of these modification techniques on the extraction of antioxidants and the low energy consumption and treatment time, more studies need to be conducted to establish them as alternative methods for improving DF properties.

Only a few studies with different results were found regarding the effect of physical and enzymatic treatments on the fermentability of fiber-rich by-products. For instance, a recent study applied a combined treatment of US and HPH to extract *Rosa roxburghii* Tratt fruit pomace IDF and improve its prebiotic potential [118]. Treated pomace IDF exhibited higher WHC and OHC, slowest fermentation rate, and highest concentrations of butyrate, an SCFA with several beneficial physiological effects. Results were related to the higher TDF content and lower particle size of the high-pressure-treated pomace. However, the impact of treatment on microbial community composition and SCFA concentrations were very low, and only slight increases of *Ruminococcus*, *Coprococcus*, and acetate production after 24 h of fermentation were detected. Tejada-Ortigoza et al. [119] investigated the impact of extrusion on the fermentation of peel by-products from orange, mango, and prickly pear. All fruit residues contained higher SDF after extrusion, which resulted in higher gas production in prickly pear peel and enhanced total SCFA production in mango and prickly pear peel. No impact of the treatment was found in orange peel; gas production and SCFA concentrations, including propionate and butyrate, even decreased. These results suggest that fermentation of DF is not only dependent on the solubility of DF, and the application of treatments inducing the solubilization of IDF does not necessarily lead to the acceleration of fermentation and higher concentrations of metabolites. Fruit and vegetable peel and pomace are a complex substrate to be hydrolyzed by gut microbiota due to their branched chemical structure, which limits fermentation. Therefore, research is more focused on the fermentation of purified oligosaccharides obtained from fruit residues and isolated DF. The application of microwave on pure IDF extracted from fruits [120] or a combination of microwave and enzymatic treatment on rice bran IDF [92] led to higher SCFAs production and shift of bacterial communities, due to the enhanced accessibility of DF for bacterial degradation.

## 4. Challenges—Predictability of the Health-Promoting Effect In Vivo

It is questionable if the determination of the in vitro capacities can serve as a tool to predict the health benefit in human bodies. Most of the studies on DF’s health-promoting properties and on antioxidant activity are tested in vitro but not in vivo. The human gastrointestinal system provides a complex environment with fluctuating conditions during the digestion process, such as dilution during mastication, salivation in the mouth and intestinal peristalsis, the presence of enzymes and different primary and secondary bile acids, and varying pH, temperature, and ionic strength due to the presence of electrolytes [121], which could highly impact the interaction of DF with the compounds of interest. It was reported that the pH of the intestinal environment differs in a fasted state between 1 and 3 in the gastric system and increases due to the duodenal secretion of alkaline bicarbonate from 6 in the duodenum to 7–8 in the terminal ileum. The intake of food increases the pH values and results in even higher pH fluctuations [122]. However, there are studies trying to simulate the gastrointestinal environment during gastric and intestinal digestion phases more closely to the real conditions, for instance by adding pepsin, amylase, pancreatin and lipase, and different bile salts, and using acidic and subsequently neutral pH [65,84,103]. It would be important to conduct studies on the impact of adapting the digestive conditions on in vitro determination.

The beneficial effect of DF intake on reducing blood total and LDL cholesterol, triglyceride, and glucose levels and preventing hyperglycemia and hypercholesterolemia in vivo is established [15]. Particularly, the enhanced excretion of bile acids, and the associated cholesterol-lowering effect, from a diet containing high levels of viscous, soluble fiber has been proven in animal and human studies [123,124]. However, the effect and the linkage between the findings of physicochemical, in vitro functional, and physiological properties of a certain fiber product with experiments in vivo are scarce. Zheng et al. [125] compared the in vitro glucose adsorption, retardation, and enzyme inhibition of TDF, IDF, and SDF fractions of bamboo shoot shell with in vivo blood glucose insulin level and oral glucose tolerance of diabetic mice, fed with a basal diet containing 5% (*w*/*w*) of these fibers for four weeks. All fiber fractions improved blood glucose level and oral glucose tolerance and, particularly, SDF with the highest GAC significantly improved blood insulin level. According to these results, the determination of GAC in vitro could be used as an instrument to estimate the antidiabetic potential in vivo. In another study, Liu et al. [126] observed that SDF of soy hulls showed higher CAC and BAC, caused by higher solubility and viscosity, than oat β-glucan and cellulose. The higher in vitro hypocholesterolemic effect was confirmed in vivo by a higher reduction of total serum cholesterol and LDL level in rats after feeding them a diet containing 4% fiber. In contrast, the in vitro results of IDF extracted from pomelo fruitlets showed higher efficacy in slowing down glucose diffusion and inhibiting α-amylase activity than SDF. However, glucose tolerance, blood glucose, and serum insulin control measured in vivo were better when feeding SDF, which does not confirm the transfer of in vitro results to the expected health-promoting effect in vivo [127]. Regarding the consistency of in vitro and in vivo fermentation experiments, studies show that the findings of in vitro fermentation might be transferable to in vivo studies. For instance, alginate oligosaccharides promoted the production of butyrate and acetate and increased the levels of *Bacteroides*, *Anaerosptipes*, and *Blautia* in vivo and in vitro [128]. Liu et al. [129] evaluated the impact of polyphenols of carrot DF with in vitro and in vivo fermentations and found the same tendency of lower total SCFA concentration and higher pH in dephenolized DF but deviations regarding the impact on SCFA profile and change of microbial communities. These discrepancies might be the result of utilizing different inocula for in vitro experiments, lower functional stability of gut microbiota in in vitro fermentation models, and lower reproducibility of results obtained with in vivo studies. Additionally, host intestinal function, such as transit time, can only be partially simulated in vitro [130].

When testing different treatment methods and conditions, in vitro methods, preferably with conditions simulating closely the digestion process, can provide the necessary information about how DF reacts to a certain modification technique to adapt the parameters. However, when the optimal treatment conditions are found, the mechanism of action, antidiabetic potential, hypocholesterolemic effect, and modulation of microbiota need to be confirmed in vivo.

## 5. Conclusions

Residues of fruit and vegetable processing are valuable sources of DF, with high potential to be used as novel functional food ingredients. Adsorption and retardation capacities of glucose, cholesterol, and bile acids, as well as α-amylase inhibition rate, serve as parameters to evaluate the antidiabetic potential and hypocholesterolemic effect in vitro. Although these health-promoting properties are the result of a complex interaction between physicochemical and functional characteristics, some of them exert a higher impact. For instance, high retardation capacities and α-amylase inhibition are associated with enhanced solubility and hydration properties, whereas binding capacities highly depend on the availability of different chemical groups, such as polyphenolic hydrophobic groups or lignin, which contribute to the interaction of DF with the target components on a molecular level. An efficient tool to achieve these desired structural alterations is the application of emergent processing technologies, emphasizing US, extrusion, and high-pressure processing. In particular, fruit and vegetable residues with high TDF content but low SDF might see improvement to their functional and health-related properties by applying these physical treatments since they cause enhanced solubility and favorable structural modification due to the exposure to high mechanical shear forces. On the contrary, enzyme application and ultrafine grinding might be the preferential method when DF with high SDF is needed and the improvement of techno-functional properties, such as thickening and texture modulation of the product, are not desired. Additionally, these methods are very efficient, safe, and easy to handle, making them suitable for large-scale implementation. However, the utilization of pure enzymes and specific strains for hydrolysis or fermentation could involve significant costs. Physical treatments, particularly high-pressure processing, have certain limitations regarding the volume and concentration of DF that can be processed in one cycle, as well as in precise control and the advanced equipment that is required. Therefore, their industrial application might encounter challenges in terms of economic feasibility. Nevertheless, the selection of treatment needs to be driven by the purpose of the ingredient’s application, such as improving the content of bioactive compounds and health benefit or ensuring storage stability. Additionally, treatment conditions must be adapted carefully to each DF material with a focus on optimizing the proportion of the SDF fraction.

Novel physical technologies have demonstrated great potential to be used as environmentally friendly methods to enable the utilization of fiber-rich fruit and vegetable residues and improve their capacity to lower the risk of developing type 2 diabetes and hypercholesterolemia. Further studies need to be conducted on using different treatment conditions and DF sources to assess the potential of each technology on a certain by-product and on studying the reducing effect of the DF product in vivo.

## Figures and Tables

**Figure 1 foods-12-03720-f001:**
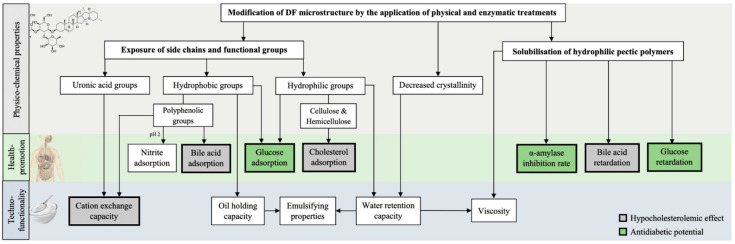
Scheme illustrating the physicochemical modifications of DF induced by the application of novel technologies (i.e., US, HPP, extrusion, microwave, or enzymes) and their impact on improving the health-promoting and techno-functional properties; arrows indicate a positive correlation/improvement of a certain health-related or techno-functional property by the structural alteration.

**Figure 2 foods-12-03720-f002:**
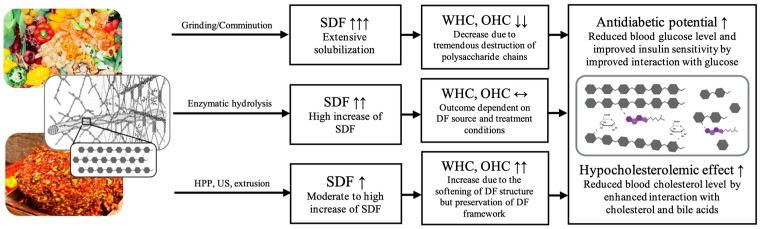
Evaluation of different studies, summarized in Table 2, of grinding, enzymatic hydrolysis, HPP, US, and extrusion regarding their expected outcome on physicochemical (SDF content), techno-functional (WHC, OHC), and health-promoting properties (glucose and cholesterol reduction). The drawing illustrates the entangled DF matrix of plant cell wall polysaccharides (adapted from Carpita and Gibeaut [99]) and highlights the packed molecular structure of cellulose and its destruction and part-solubilization after treatments, which led to facilitation of the adsorption of molecules, such as glucose and cholesterol.

**Table 1 foods-12-03720-t001:** Potential fruit and vegetable DF sources based on the proportion of waste, their DF composition, and total phenolic content.

DF Source	Global Production (MMT)	Waste (% of Fruit Weight)	Fiber Composition (%)			Total Phenolic Content (A—mg GAE/g DM; B—mg GAE/g Peel/Pomace)
			TDF	IDF	SDF	
Apple	87.24	20–40	**Pomace**			
			76.84 78.20 60.10	57.87 63.90 46.30	18.97 [22] 14.33 [8] 13.80 [11]	4.80 [9] **A**
			**Peel**			
			47.80 35.22	42.10 28.73	5.80 [23] 6.48 [24]	38.60 [23] **A** 5.00–5.88 * [25] **B**
Orange	78.7	50	**Pomace**			
			54.82	29.65	25.17 [22]	8.62 [9] **A**
			**Peel**			
			57.00	47.60	9.41 [26]	2.84 [27] **B**
Lemon, Lime	20.05	50	**Pomace**			
			77.93 74.94	50.02 46.93	27.91 [28] 28.10 [29]	13.79 [30] **A**
			**Peel**			
			53.02 82.14	34.54 54.01	18.48 [28] 28.01 [29]	2.23–3.63 * [27] **B**
Mango	55.85	45	**Pomace**			
			28.05	13.80	14.25 [31]	16.1 [31] **A**
			Peel			
			44.70–78.40 *	28.99–50.33 *	15.70–28.06 * [32]	54.67–109.07 * [32] **B**
Peach	25.74	-	**Pomace**			
			36.10 54.20	23.80 35.40	12.30 [33] 19.10 [34]	0.84 [35] **B**
Carrot	44.76	30–50	**Pomace**			
			63.60 52.00	50.10 42.10	13.50 [36] 9.91 [24]	0.02–0.39 * [37] **A**
Onion	99.97	-	**Bottom tissue**			
			40.80	30.60	10.70 [38]	-
			**Skin**			
			68.30	66.60	1.70 [38]	69.23 [39] **A**

**TDF**—total DF content (TDF = IDF + SDF), **IDF**—insoluble DF content, **SDF**—soluble DF content, **GAE**—gallic acid equivalents, **DM**—dry matter; ***** values of different fruit cultivars and ripening grades measured in the indicated study, the minimum and maximum values are stated.

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
