# Peer review of "Relationship between Physicochemical, Techno-Functional and Health-Promoting Properties of Fiber-Rich Fruit and Vegetable By-Products and Their Enhancement by Emerging Technologies"

_foods, 2023, doi:10.3390/foods12203720_

Round 1
Reviewer 1 Report (Previous Reviewer 1)
Title of “Interaction of structural, techno-functional and health-promoting properties of fiber-rich fruit and vegetable by-products and enhancement through physical and enzymatic treatments” is inappropriate.
At first, although you insist “We would like to point out again that the objective of this article was not to give a detailed overview of the transformation of DF by traditional and novel procedures although a section was dedicated to this aspect.”, title contains “enhancement through physical and enzymatic treatments”. Thus, normally, readers think this manuscript focusing on not only “Interaction of structural, techno-functional and health-promoting properties of fiber-rich fruit and vegetable by-products” but also “enhancement through physical and enzymatic treatments”. If it does not contain explanation of “enhancement through physical and enzymatic treatments” in your objective of this manuscript, it must delete sentence related with physical and enzymatic treatments in title and section of Introduction. Additionally, it must delete section of “3. Enhancing the health-promoting effect of DF by applying novel technologies” because the authors insist that the objective of this article was not to give a detailed overview of the transformation of DF by traditional and novel procedures. Finally, I don’t care your manuscript contains “enhancement through physical and enzymatic treatments” such as section of “3. Enhancing the health-promoting effect of DF by applying novel technologies” or not. However, if title and Introduction refers, the manuscript must require detail of these contents since this manuscript is review type.
At second, maybe, the authors misunderstand a mean of “interaction”. Interaction means that one thing affects other thing and other thing affects one thing. I can understand effect of structural or techno-functional properties of dietary fiber on health-promoting properties. However, I cannot understand effect of health-promoting properties on structural or techno-functional properties of dietary fiber. Actually, in the manuscript for section of “Interaction between physicochemical, techno-functional and health-promoting properties of DF”, it describes only effect of structure or chemical characteristic on health benefit properties of DF. Conclusively, in my opinion, it is better to revise this manuscript title “Effect of structure and function of processed fiber-rich agriproduct waste substance on health-promoting properties”.
Although my former reject was decided by why the logical structure is not enough in this manuscript, this manuscript has a lack of logical structure still. Therefore, Re-submit must be conducted after drastic revising whole of manuscript.
Author Response
Dear reviewer,
We appreciate your opinion and comments. As you mentioned, the phrase ‘Interaction’ could be misinterpreted since we only describe the relationship between physicochemical and techno-functional properties and their impact on health-related aspects of fiber-rich by-products. Therefore, the title was revised to ‘Relationship between physicochemical, techno-functional and health-promoting properties of fiber-rich fruit and vegetable by-products and their enhancement by emerging technologies’ and the caption of section 2 to ‘Impact of physicochemical properties of DF on its techno-functional and health-promoting characteristics’ (modifications are marked in red). By changing physical and enzymatic treatments to emerging technologies in the title, our intention is to reduce the focus on the treatments and their mechanisms. With this adjustment, we hope that readers will expect less detailed information for each treatment method but a general overview about the correlation among DF properties and potential enhancements.
Yours sincerely,
Robert Soliva-Fortuny
Reviewer 2 Report (Previous Reviewer 4)
Accept in current format
Author Response
Dear reviewer,
We are grateful for accepting our manuscript for publication and appreciate the valuable feedback provided previously.
Yours sincerely,
Robert Soliva-Fortuny
Reviewer 3 Report (Previous Reviewer 2)
Dear Authors,
I find revisions you have made according to my comments satisfactory, although I strongly recommend that the economic feasibility aspect be considered more thoroughly in the future. I find this paper now suitable for publication.
I found the English is good enough, and only minor edits, if any, are needed.
Author Response
Dear reviewer,
We are grateful for accepting our manuscript for publication and appreciate the valuable feedback provided previously.
Yours sincerely,
Robert Soliva-Fortuny
Reviewer 4 Report (Previous Reviewer 3)
Dear Authors
The suggestions made throughout the manuscript have been answered. Favorable opinion for publication.
Author Response
Dear reviewer,
We are grateful for accepting our manuscript for publication and appreciate the valuable feedback provided previously.
Yours sincerely,
Robert Soliva-Fortuny
This manuscript is a resubmission of an earlier submission. The following is a list of the peer review reports and author responses from that submission.
Round 1
Reviewer 1 Report
Major comment
This review type of manuscript describes a dietary fiber on healthy merit and processing properties. This review is good information for sustainable development technology. However, in my opinion, this manuscript is too many information for review type of manuscript to be consistent properties. At least, it must separate the sentences on healthy merit and processing properties of a dietary fiber. The gap between the research fields is too wide. In the result of the gap, description for each of the research fields become insufficiently. In my honest advice, it is better for the manuscript to focus on processing properties.
Minor comment
Checking whole manuscript
It was not applied superscript on whole of reference numbers in manuscript.
(ex. L45, L49, L54)
L29-73
In my honest advice for “Introduction”, it is better to described Sustainable Development Goals (SDGs) related with food wastes and human health.
Additionally, in end of “Introduction”, it is better to describe structure of this review briefly.
L50
Mean of “bioactive compounds” is unclear.
L71
Mean of “potential environmental problems” is unclear.
L76
Mean of “fruits and vegetable by-products” is unclear.
Maybe, it is assumed that “fruits and vegetable by-products” means by-products in post-processing fruits and vegetable.
However, mean of “fruits and vegetable by-products” contains by-products before and in and after harvest.
L100-103
Considering also the high availability of pectic polysaccharides, namely 17.31% in apple and 12.38% in orange pomace, and their high phenolic content, both can be selected as sources with high potential.
Where is it noted percentage value such as 17.31% in apple and 12.38% in orange pomace in Table 1?
Additionally, an empty space between “percentage value” and “%” is required such as “12.38 %”.
L106-108
Lime peel and apple pomace are mainly used for the production of industrial pectin but mango peel and peach pomace show high potential to be used as additional sources.
The reason is unclear.
Why has it been mainly used lime peel and apple pomace for the production of industrial pectin?
Required quality and component of vegetable and fruits for the production of industrial pectin should be described.
L134-137
DF are non-digestible polysaccharides which leads to the fermentation by intestinal microbiota inside the colon and causes a pH reduction by the production of short chain fatty acids (SCFA) (e.g., butyrate, propionate, acetate).
In my opinion, it is better to revise “non-digestible polysaccharides which leads to the fermentation by intestinal microbiota inside the colon” to “non-digestible polysaccharides which is one of the substrates for fermentation by intestinal microbiota inside the colon”.
Additionally, it is better to revise “a pH reduction” to “acidifying the colon inside”.
L352
Means of “a greater extent” is unclear.
L353-354
Means of “accessibility of insoluble DF” is unclear.
L364-366
In addition, the presence of polyphenolic components, which are highly present in fruit by-products, is associated with strong antibacterial effect and the stimulation of beneficial bacterial species increasing fermentability.
It is seemed that strong antibacterial effect by polyphenolic components is conflicted with the stimulation of beneficial bacterial species increasing fermentability.
More description about these is required.
L381-
It must be described effect of preparation treatment before altering DF on quality and components of final DF production.
Or, does preparation treatment such as wet/dry milling and dehydration have no effect on quality and components of final DF production?
L367-488
This research paper is manuscript of review type.
Thus, firstly, fundamental mechanism for changing DF components must be described.
For example, it is a mechanism for heating transition from high molecule DF to low molecule DF.
Additionally, these sentences refer a novel technology for changing DF components.
However, no description for actual experimental result in previous study is shown.
Only a general description was shown in these sentences.
At least, description for the actual results or assumption in the other’s or author’s researches required for manuscript of review type.
L415-417
the breakage of cell walls, facilitated mass transfer of compounds and degradation and change of the molecular structure of polysaccharides
It must be described why it is important for improving the functionality of processed DF to lead the breakage of cell walls, facilitated mass transfer of compounds.
Additionally, it must be described detail of a chemical change for the molecular structure of polysaccharides treated by US.
L489-642
It should be combined description of an actual experimental result in previous study at chapter of 5.3 with a general description at chapter of 5.2 for the reader’s good understanding.
L553-554
Detail of “the results of techno-functional and health-re-553 lated properties were not consistent” must be described.
Table 1
“TDF IDF SDF” in top of table 1 should be revised to “IDF + SDF = TDF”.
Additionally, size of table 1 should be minimized to 1 page size.
Mean of “GAE” and “DM” is unclear.
Please read minor comment
Author Response
Dear reviewer,
We are submitting a revised version of our review. Modifications can be found in red throughout the manuscript.  We appreciate your comments and suggestions. We are providing a list of the changes carried out in response to the recommendations. 
Yours sincerely,
Robert Soliva-Fortuny
This review type of manuscript describes a dietary fiber on healthy merit and processing properties. This review is good information for sustainable development technology. However, in my opinion, this manuscript is too many information for review type of manuscript to be consistent properties. At least, it must separate the sentences on healthy merit and processing properties of a dietary fiber. The gap between the research fields is too wide. In the result of the gap, description for each of the research fields become insufficiently. In my honest advice, it is better for the manuscript to focus on processing properties.
The objective of the review was to give an overview about the application of the most studied physical and enzymatic treatments on fruit and vegetable by-products and the suspected outcome on different properties of DF. In addition, it was of high interest to explain the interaction of the structural, functional and health-related properties since, to the authors’ knowledge, currently no articles are available summarizing the different findings regarding this complex relationship. The application of DF into food products is mainly driven by the objective to enhance the nutritional value of the product. Therefore, studies were selected which investigated the impact of different structural and techno-functional characteristics of DF in relationship with the health outcome.
It was not applied superscript on whole of reference numbers in manuscript.
We have carefully revised the manuscript regarding the format of the journal, references and language.
L29-73: In my honest advice for “Introduction”, it is better to described Sustainable Development Goals (SDGs) related with food wastes and human health. Additionally, in end of “Introduction”, it is better to describe structure of this review briefly.
The introduction of the review was restructured including a description of the structure, as suggested. Sustainable Development Goals are mentioned in the revised manuscript.
L50: Mean of “bioactive compounds” is unclear.
“Bioactive compounds” were changed to “phytochemicals”. The importance of phytochemicals/antioxidant compounds is further explained in L51ff.
L71: Mean of “potential environmental problems” is unclear.
Since the introduction was restructured, “potential environmental problems” now clearly refer to the underutilization of fruit and vegetable by-products (L35-37).
L76: Mean of “fruits and vegetable by-products” is unclear. Maybe, it is assumed that “fruits and vegetable by-products” means by-products in post-processing fruits and vegetable. However, mean of “fruits and vegetable by-products” contains by-products before and in and after harvest.
As assumed, “fruits and vegetable by-products” refer to post-processing by-products mainly from the juice processing industry.
According to that, it was added: “When selecting fruits and vegetable by-products, primarily obtained from juice preparation and processing, as novel sources of DF, several factors should be considered.”
L100-103: Considering also the high availability of pectic polysaccharides, namely 17.31% in apple and 12.38% in orange pomace, and their high phenolic content, both can be selected as sources with high potential. Where is it noted percentage value such as 17.31% in apple and 12.38% in orange pomace in Table 1? Additionally, an empty space between “percentage value” and “%” is required such as “12.38 %”.
All percentages were revised and empty spaces added. In Tab. 1 the SDF contents of the different sources were summarized. The percentages (17.31 % in apple and 12.38 % in orange) only refer to the pectin content which is the major part of SDF. According to that, it was revised: “Considering also the high SDF content, therein high availability of pectic polysaccharides (17.31 % in apple and 12.38 % in orange pomace) and their high phenolic content, both can be selected as sources with high potential.”
L106-108: Lime peel and apple pomace are mainly used for the production of industrial pectin but mango peel and peach pomace show high potential to be used as additional sources. The reason is unclear. Why has it been mainly used lime peel and apple pomace for the production of industrial pectin? Required quality and component of vegetable and fruits for the production of industrial pectin should be described.
Reasons why lime peel and apple pomace are commonly used as pectin source were added: “Lime peel and apple pomace are mainly used for the production of industrial pectin mainly due to the abundant availability of the raw material and enhanced accessibility of pectin with high molecular weight, high degree of esterification and good gelling properties [17].”
L134-137: DF are non-digestible polysaccharides which leads to the fermentation by intestinal microbiota inside the colon and causes a pH reduction by the production of short chain fatty acids (SCFA) (e.g., butyrate, propionate, acetate). In my opinion, it is better to revise “non-digestible polysaccharides which leads to the fermentation by intestinal microbiota inside the colon” to “non-digestible polysaccharides which is one of the substrates for fermentation by intestinal microbiota inside the colon”. Additionally, it is better to revise “a pH reduction” to “acidifying the colon inside”.
Changes were made according to the reviewer’s suggestions: “DF are non-digestible polysaccharides which are substrates for fermentation by intestinal microbiota inside the colon. During the metabolism of DF by gut bacteria, short chain fatty acids (SCFA) (e.g., butyrate, propionate, acetate) are produced leading to colonic acidification which inhibits the growth of pathogenic organisms and carcinogens and thereby reduces the risk of colon carcinogenesis [28].”
L352: Means of “a greater extent” is unclear.
L353-354: Means of “accessibility of insoluble DF” is unclear.
Both sentences were rewritten in order to clarify. “The uptake and metabolization of DF by gut microbiota species are strongly connected with DF structure whereas soluble DF is more easily fermentable than insoluble DF [6]. This is mainly due to the lower degradability of insoluble DF for bacterial polysaccharide hydrolases based on their lower accessible, more dense and cross-linked cell wall structure [64] and the enhanced metabolization of shortened DF fractions of low DP, such as cello- xylo and pectin oligosaccharides [65].”
L364-366: In addition, the presence of polyphenolic components, which are highly present in fruit by-products, is associated with strong antibacterial effect and the stimulation of beneficial bacterial species increasing fermentability. It is seemed that strong antibacterial effect by polyphenolic components is conflicted with the stimulation of beneficial bacterial species increasing fermentability. More description about these is required.
The conflict was resolved by explaining that the antibacterial effect is only associated with the inhibited growth of certain pathogenic species, such as H. pylori. Several studies report the inhibition of some pathogens but at the same time the promotion of the growth of beneficial bacteria.
L381- It must be described effect of preparation treatment before altering DF on quality and components of final DF production. Or, does preparation treatment such as wet/dry milling and dehydration have no effect on quality and components of final DF production?
It was added that parameters of the preparation method, such as the drying temperature and the particle size obtained after milling, already influence the characteristics of the DF product.
L367-488: This research paper is manuscript of review type. Thus, firstly, fundamental mechanism for changing DF components must be described. For example, it is a mechanism for heating transition from high molecule DF to low molecule DF. Additionally, these sentences refer a novel technology for changing DF components. However, no description for actual experimental result in previous study is shown. Only a general description was shown in these sentences. At least, description for the actual results or assumption in the other’s or author’s researches required for manuscript of review type.
The principle of physical treatments of opening the DF structure by mechanical shear forces and inducing solubilization was added. Results of the application of these treatments and outcomes are described in the next sections (5.2). 5.1 only gives a short summary of the general mechanism and the parameters which need to be considered when applying a certain technology.
L415-417: the breakage of cell walls, facilitated mass transfer of compounds and degradation and change of the molecular structure of polysaccharides It must be described why it is important for improving the functionality of processed DF to lead the breakage of cell walls, facilitated mass transfer of compounds. Additionally, it must be described detail of a chemical change for the molecular structure of polysaccharides treated by US.
Cell breakage leads to solubilization of DF which, as mentioned several times in the articles, is of high importance for the improvement of several DF properties. To clarify, it was revised: “This microscale implosion induces high shear forces, localized increase of temperature and pressure and the generation of hydroxyl radicals which can lead to the breakage of cell wall polysaccharides and their degradation to shorter DF fractions. Additionally, the process facilitates the release of intracellular components.” The chemical change of the molecular structure depends on several factors, such as the treatment conditions and chemical nature of the used substrate. Therefore, it is very difficult to give a general description of the chemical change of the microstructure. Furthermore, it was decided not to give further details of the mechanism of US since the objective of this section was only to give a brief description of all treatments applied in the studies in section 5.2.
L489-642: It should be combined description of an actual experimental result in previous study at chapter of 5.3 with a general description at chapter of 5.2 for the reader’s good understanding.
Chapter 5.3 is related with 5.2 since also studies on the application of different modification technologies are presented. However, the context is different since in 5.3 only studies were selected which applied in vitro and in vivo experiments in order to evaluate their transferability. Chapter 5.2 only focuses on in vitro studies and the impact of modification technologies on different DF properties. Hence, the separation of these sections is important for the understanding.
L553-554: Detail of “the results of techno-functional and health-related properties were not consistent” must be described.
In this part, the higher expected efficiency of physical treatments in techno-functional improvement was emphasized since the results regarding the enhancement of techno-functional properties after EH were not consistent. Changes for clarifications were made: “Several studies on different DF sources confirm the high efficiency of enzymes to increase the SDF content but the results regarding the impact on techno-functional properties were not consistent (Tab. 2). However, in most of the studies, higher SDF content induced by enzymatic hydrolysis led to the promotion of health-related adsorption and retardation capacities [46,50,51,85].”
Table 1: “TDF IDF SDF” in top of table 1 should be revised to “IDF + SDF = TDF”. Additionally, size of table 1 should be minimized to 1 page size. Mean of “GAE” and “DM” is unclear.
TDF, GAE and DM explanations were added below the table. The size of the table was significantly reduced.
Reviewer 2 Report
The abstract effectively introduces the main topic of the paper.
The conclusion effectively summarizes the main points discussed in the paper.
The paper is very well structured and provides a clear overview of previous research. A particular strength of the paper is that it also presents an overview of challenges related to predictability of the health-promoting effect in vivo.
The paper covers all the important novel physical, environmentally friendly technologies such as ultrasound, high-pressure processing, extrusion, and microwave. The mention of the application of enzymes and their impact on soluble DF content is interesting.
The only minor critique of the paper, and perhaps a suggestion for improvement, is to briefly discuss in one or two sentences (maybe in conclusion) which of these mentioned techniques would have the greatest potential for commercialization, considering economic feasibility.
Additionally, there are some minor technical issues such as different fonts (sizes) within the paper itself, which I believe will be addressed during the technical review process.
If possible, please replace the figure with higher-resolution image (Text is not clearly visible)
In conclusion, this is a paper of exceptional quality, and with very minor revisions.
The paper will make an exceptional scientific contribution as it effectively summarizes previous research and provides recommendations for future research directions.
Author Response
Dear reviewer,
We are submitting a revised version of our review. Modifications can be found in red throughout the manuscript.  We highly appreciate your comments and suggestions. Changes were carried out in response to the recommendations:
- A short discussion at the end of the manuscript regarding the commercialization and economic feasibility of novel physical technologies and enzymatic treatments was added
- The manuscript was carefully revised regarding the format of the references and different fonts
- Fig. 2 was replaced with a higher-resolution image
Yours sincerely,
Robert Soliva-Fortuny
Reviewer 3 Report
Dear Authors
the manuscript entitled "Enhancement of structural, techno-functional and health-related properties of fiber-rich fruit and vegetable by-products through physical and enzymatic treatments" is well written, minor corrective suggestions have been made throughout the attached text.

Author Response
Dear reviewer,
We are submitting a revised version of our review. Modifications can be found in red throughout the manuscript.  We highly appreciate the comments and suggestions. We are providing a list of the changes carried out in response to the recommendations. 
Yours sincerely,
Robert Soliva-Fortuny
- Page 1: A more recent reference was added (FAO 2019).
- Page 2: References were adapted to the format of the journal.
- Page 3: Table 1 was revised for the reader’s better understanding, explanations for the abbreviations of GAE and DM were added. Different values, stated for the DF composition or TPC, refer to the different cultivars and ripening grades of fruit or vegetables which were tested in the same study. This was explained below the table and the corresponding values marked with *.
- Page 11: The title of Table 2 was shortened, and explanations and abbreviations were placed below the table.
- Page 16: The scientific name was changed to italic font.
Reviewer 4 Report
This review explored an interesting question that is to identify the effects of physical and enzymatic treatments on fiber-rich byproducts to enhance the health-promoting properties. Overall, the aim of the study is interesting, there are some key questions that needs to be addressed in the revised manuscript:
1. Regarding the experimental design: Why the authors included these fruits and vegetables? What was the inclusion criteria? It is recommended to include a brief description of the methodology, what were the criteria for the selection of the articles.
2. The principal question is the logical sequence of the article. It is recommended to rearrange the ideas in the introduction and other sections. For example, what is the problem to be studied? Low fiber intake or the amount of food by-products? The next point could be the properties of the fiber and after that what about techniques to improve fiber digestibility?
3. The digestibility depends on the food source?
Minor comments
Improve tables and figures,
The format of the references
The sequence of citations in the document
The font
Author Response
Dear reviewer,
We are submitting a revised version of our review. Modifications can be found in red throughout the manuscript.  We appreciate your comments and suggestions. We are providing a list of the changes carried out in response to the recommendations. 
Yours sincerely,
Robert Soliva-Fortuny
- Regarding the experimental design: Why the authors included these fruits and vegetables? What was the inclusion criteria? It is recommended to include a brief description of the methodology, what were the criteria for the selection of the articles.
The inclusion criteria of fruit and vegetable by-products which might have high potential to be used as novel food ingredients are revised at the beginning of section 2:
“When selecting fruits and vegetable by-products, primarily obtained after juice preparation and processing, as novel sources of DF, several factors should be considered. These include the content of SDF and bioactive compounds present in the by-products as well as the global production of the specific fruit or vegetable and the amount of by-product generated after processing. In Tab. 1 fruit and vegetables and their by-products with high potential to be used as novel ingredients are collected focusing on the amount of by-product which is produced per year and the DF composition including SDF and total phenolic content (TPC).”
Additionally, a summary of the structure of the article was added at the end of the introduction explaining that articles which applied treatments on novel DF substrates/fiber-rich substrates and investigated the outcome on several properties of DF, i.e. physicochemical, techno-functional and health-related characteristics, were included.
- The principal question is the logical sequence of the article. It is recommended to rearrange the ideas in the introduction and other sections. For example, what is the problem to be studied? Low fiber intake or the amount of food by-products? The next point could be the properties of the fiber and after that what about techniques to improve fiber digestibility?
The introduction was restructured to clarify that primarily the high amount of fruit and vegetable by-products which poses a threat to the environment is the main problem which requires a solution. However, since these by-products contain a high DF content and the average intake of DF within most populations is lower than the recommended, the incorporation into food is of high interest and would address another problem. Afterwards, DF properties and the advantage of high SDF for sensorial, techno-functional and health-promoting properties, including fermentability, are presented.
Studies regarding the studies which applied different techniques to improve SDF content and fermentability are summarized in L615-642 (end of section 5.2).
- The digestibility depends on the food source?
As explained in section 4.2.3., digestibility mainly depends on DF structure and thereby on the chemical nature of the DF substrate. The section was revised aimed at improving comprehension.
Minor comments: Improve tables and figures, the format of the references, the sequence of citations in the document, the font
Some format problems, such as different fonts or replaced values in Tab. 1, were not in the original draft and must be due to the change to the journal’s format. The format of the references was adapted to the journal’s format. Tab. 1, Fig. 1 and Fig. 2 were improved for the reader’s better understanding. Citations were carefully revised, and 26 references were removed.
Reviewer 5 Report
This review presents a theme of great interest and with a great collection of information. However, it focuses mainly on the biological aspects of the effect of dietary fiber modification and less in relation to the technological aspects in food. Evaluate the title.
On the other hand, I recommend rearranging the order of the topics covered in the text. Attached file with comments and observations

Author Response
Dear reviewer,
We are submitting a revised version of our review. Modifications can be found in red throughout the manuscript.  We appreciate your comments and suggestions. We are providing a list of the changes carried out in response to the recommendations. 
Yours sincerely,
Robert Soliva-Fortuny
Some format problems, such as the higher letter sizes, replaced values in Tab. 1 or misplacement of Tab. 2, were not in the original draft and must be due to the change to the journal’s format. We have carefully revised the manuscript regarding the journal’s format of the references and language.
- Page 1
- Chemical methods were included as less efficient techniques in the abstract. The limit of words did not allow to explain the advantages of physical methods further.
- Page 2
- Chronic diseases were changed to noncommunicable diseases.
- The SDF content of more than 30% was suggested in reference [11]. This minimum is related to the advantageous properties of SDF in terms of technological, nutritional and sensorial aspects which are explained in the following sentences. A reference was added for the enhancing impact of SDF on fermentability, blood and cholesterol reduction. Solubility was referred to DF substrates: “DF high in soluble polysaccharides and polyphenols exhibits enhanced fermentability and ability to reduce blood glucose and cholesterol [7]. Furthermore, high solubility of a fiber-rich substrate is associated with favourable technological functionality, such as the capacity of forming gels, viscosity increase, emulsion stability and water/oil binding capacity [8] as well as improved sensorial properties, such as reduced graininess and chewiness [12].”
- The objective of increasing SDF and the impact on techno-functionality and health-related properties is discussed in detail in the next sections of the article.
- To the authors’ knowledge, there are no regulations regarding the maximum of SDF for its nutritional effect.
- TPC was exchanged in bioactive compounds.
- Page 3/Tab.1
- The whole table was revised according to the reviewer’s suggestions.
- References for each fiber composition and TPC are stated as superscript numbers behind the values.
- The potential of these DF sources/by-products as novel ingredients in the title refer to “the content of SDF and bioactive compounds present in the by-products as well as the global production of the specific fruit or vegetable and the amount of by-product generated after processing.”, explained at the beginning of section 2.
- The * refer to the different values of DF and TPC of different fruit varieties and ripening grades which were tested in one study. This was explained at the bottom of the table.
- Page 4
- The percentages (17.31 % in apple and 12.38 % in orange) only refer to the pectin contents of the pomaces as the major part of SDF. According to that, it was revised: “Considering also the high SDF content, therein high availability of pectic polysaccharides (17.31 % in apple and 12.38 % in orange pomace) and their high phenolic content, both can be selected as sources with high potential.”
- In section 3 only the general health benefit of DF was focused without separation in IDF and SDF. The advantage of high SDF for the health outcome was already mentioned in the introduction and the effect of microstructure alterations and enhanced SDF is discussed in detail in section 4.2 and 5.2.
- Page 5 to 8
- The daily recommended intake of 25-35 g/day of DF is mentioned in the introduction.
- Insulating myelin was changed to “neuron’s myelin sheaths”.
- The title of 4 was changed to clarify that the relationship between the physicochemical with the techno-functional and health-promoting properties of DF is discussed.
- The objective of section 4 was to explain the interaction of physicochemical, techno-functional and health-promoting properties without the application of modification technologies. This complex relationship was illustrated in Fig. 2. In the following sections (5.2), this knowledge was used to explain the findings of different studies applying treatments on DF.
- Page 9
- The meaning of conventional methods was clarified by revising L381-385: “The high energy and time consumption of traditional physical treatments (i.e. grinding, high-speed mixing and steam explosion), potential damage of DF structure and degradation of bioactive compounds after thermal technologies, and the critical environmental impact of acidic and basic solvents during chemical applications, are some of the disadvantages of these conventional methods.”
- In the conclusion a short discussion regarding the economic feasibility of the methods was added.
- Page 11
- The title of Tab. 2 was shortened, explanations and abbreviations were placed below the table.
- Fig. 2 serves as a model to explain the structural and functional characteristics which have the greatest impact on the hypercholesterolemic effect and antidiabetic potential. To the authors’ knowledge, there is currently no article available which summarizes the different findings regarding this complex relationship. Therefore, illustrations were focused to enhance the comprehension of this interaction. The different parameters which influence these health-promoting properties, explained in section 4.2, were considered as basic knowledge, and therefore do not require further illustration.
- Page 12
- There are no general efficiency percentages for the treatments since the efficiency varies between the used DF materials and the specific DF characteristics. For instance, US might be very efficient to enhance the SDF and AAIR of orange peel but does not show improvements of these properties in carrot pomace although CEC improves in this substrate. Disadvantages for the application of novel physical and enzymatic treatments were added in the conclusion.
- Page 14
- 5.1 was placed before 5.2 to explain firstly the general mechanisms of the treatments to discuss afterwards in 5.2 their application on different DF substrates and their outcomes.
- Page 15
- As stated in the title of Fig. 1, the references are the findings of Tab. 2. Studies were tried to evaluate regarding the general tendency of the impact of each treatment.
Round 2
Reviewer 1 Report
In my honest opinion, my reject decision on this manuscript is not changed even with considering author’s response. In manuscript of review type, it requires to contain content ranging from the basis to the advance and the latest for primary learner’s understanding. However, this manuscript let the readers to understand general background for transformation from postharvest waste of fruits and vegetable to dietary fiber but not to understand mechanism of the transformation. This manuscript does not contain even mechanism of general traditional technology such as enzyme reaction and hydrolysis reaction. Additionally, it is not indicated difference between traditional and novel technology for degradation of insoluble dietary fiber. In my opinion, a major revise cannot improve quality achieving manuscript of review type. Re-submit must be conducted after drastic revising whole of manuscript.
Author Response
Dear reviewer,
We highly appreciate your comments. We are resubmitting a revised version of our review with modifications marked in red throughout the manuscript.
Yours sincerely,
Robert Soliva-Fortuny
In my honest opinion, my reject decision on this manuscript is not changed even with considering author’s response. In manuscript of review type, it requires to contain content ranging from the basis to the advance and the latest for primary learner’s understanding. However, this manuscript let the readers to understand general background for transformation from postharvest waste of fruits and vegetable to dietary fiber but not to understand mechanism of the transformation. This manuscript does not contain even mechanism of general traditional technology such as enzyme reaction and hydrolysis reaction. Additionally, it is not indicated difference between traditional and novel technology for degradation of insoluble dietary fiber. In my opinion, a major revise cannot improve quality achieving manuscript of review type. Re-submit must be conducted after drastic revising whole of manuscript.
Following your suggestions, the section ‘1.3 Technologies with high potential to be applied on novel DF sources’ was drastically revised. Detailed information regarding the mechanism of traditional modification technologies (L. 192-200), novel technologies (L. 203-217) and enzymatic hydrolysis (L. 279-293) was added. Some additional facts about the mechanism of HPH and extrusion were included to improve the comprehension of the mechanism of these treatments in the transformation of DF. Additionally, the advantage of novel technologies was emphasized (L. 198-217).
Significant structural changes were implemented to enhance the comprehension of the review. After defining the objective of utilizing fiber-rich by-products to ‘overcome potential environmental problems but also help to close the gap between actual and recommended intake in the population and improve human health’, the description of the methodology and structure of the review at the beginning of the introduction (L. 47-66) was revised. The introduction was restructured containing basic information about DF properties focusing on its health benefit (1.1), fruit and vegetable by-products with potential as innovative DF sources (1.2)and the principles of physical and enzymatic treatments and the advantage of novel technologies (1.3). Afterwards, the interaction between physicochemical, techno-functional and health-promoting properties was described (2) which constitute the foundation for understanding the following section comparing several studies of applying different technologies to enhance these properties (3). The numbers of Fig. 1 and 2 were changed to follow this more coherent order. This approach aims to facilitate the distinction between fundamental knowledge and novel insights regarding the linkage between DF structure, techno-functionality and health promotion as well as methodologies to enhance certain properties. As before, the review finishes with the discussion of the transferability of the health-promoting effect measured in vitro to in vivo (4) and a general conclusion (5).
We would like to point out again that the objective of this article was not to give a detailed overview of the transformation of DF by traditional and novel procedures although a section was dedicated to this aspect. Several articles about the modification technologies of DF and fiber-rich by-products already exist, e.g. Garcia-Amezquita et al., 2018 and Gan et al., 2021. This article was aimed to summarize information about the interaction of DF properties and the importance of some structural characteristics which could be valuable when the industry needs to select a certain DF source and technology for its incorporation into a food product. Therefore, the title was changed into ‘Interaction of structural, techno-functional and health-promoting properties of fiber-rich fruit and vegetable by-products and enhancement through physical and enzymatic treatments.’
Reviewer 4 Report
As the last review report, the topic of this review is interesting, however, the key questions that were previously formulated have not been clarified. It is suggested to restructure the review, emphasizing the methodology to understand what the inclusion criteria are.
Also, it is suggested to review the logical sequence of the article.
Author Response
Dear reviewer,
We highly appreciate your comments. We are resubmitting a revised and restructured version of our review with modifications marked in red throughout the manuscript.
Yours sincerely,
Robert Soliva-Fortuny
As the last review report, the topic of this review is interesting, however, the key questions that were previously formulated have not been clarified. It is suggested to restructure the review, emphasizing the methodology to understand what the inclusion criteria are.
Also, it is suggested to review the logical sequence of the article.
Following your suggestions, significant structural changes were implemented to enhance the comprehension of the review. After defining the objective of utilizing fiber-rich by-products to ‘overcome potential environmental problems but also help to close the gap between actual and recommended intake in the population and improve human health’, the description of the methodology, inclusion criteria (L. 54-56 and 128-135) and structure of the review at the beginning of the introduction (L. 47-66) was revised. The introduction was restructured containing basic information about DF properties focusing on its health benefit (1.1), fruit and vegetable by-products with potential as innovative DF sources (1.2) and the principles of physical and enzymatic treatments and the advantage of novel technologies (1.3). Afterwards, the interaction between physicochemical, techno-functional and health-promoting properties was described (2) which constitute the foundation for understanding the following section comparing several studies of applying different technologies to enhance these properties (3). The numbers of Fig. 1 and 2 were changed to follow this more coherent order. This approach aims to facilitate the distinction between fundamental knowledge and novel insights regarding the linkage between DF structure, techno-functionality and health promotion as well as methodologies to enhance certain properties. To emphasize the objective of the review, the title was changed into ‘Interaction of structural, techno-functional and health-promoting properties of fiber-rich fruit and vegetable by-products and enhancement through physical and enzymatic treatments.’ As before, the review finishes with the discussion of the transferability of the health-promoting effect measured in vitro to in vivo (4) and a general conclusion (5).
Additionally, the section ‘1.3 Technologies with high potential to be applied on novel DF sources’ was revised and detailed information regarding the mechanism of traditional modification technologies (L. 192-200), novel technologies (L. 208-213) and enzymatic hydrolysis (L. 279-293) was added. Some additional facts about the mechanism of HPH and extrusion were included to improve the comprehension of the mechanism of these treatments in the transformation of DF.
Reviewer 5 Report
The authors incorporated the requested changes
The authors incorporated the requested changes
Author Response
Dear reviewer,
We are resubmitting a revised and restructured version of our review with modifications marked in red throughout the manuscript.
Yours sincerely,
Robert Soliva-Fortuny